# EMPIRICAL NTK TRACKS TASK COMPLEXITY

## ABSTRACT

Mathematical properties of the neural tangent kernel (NTK) have been related–both theoretically and empirically–to convergence of optimization algorithms and the ability of trained models to generalize. However, most existing theoretical results hold only in the infinite width limit and only for standard data distributions. In the present work, we suggest a practical approach to investigating the NTK for finite-width networks, by understanding the parameter space symmetries of the network in the presence of finite data sets. In particular, the NTK Gram matrix associated to any finite data set can naturally be regarded as an empirical version of the NTK. Moreover, its rank agrees with the *functional dimension* of the data set, the number of independent parameter perturbations affecting the model's outputs on the data set. In this work, we explore the evolution of the functional dimension of deep ReLU networks during training, focusing on the relationship to data set complexity, regularization, and training dynamics. Empirically, we find that functional dimension of deep ReLU networks: (1) tracks data set complexity, (2) increases during training until function stabilization, and (3) decreases with stronger weight decay, suggesting that gradient-based optimization algorithms are biased towards simpler functions for ReLU networks. Moreover, our experiments provide strong evidence that–contrary to conventional wisdom–the empirical NTK for deep finite-width ReLU networks is typically rank-deficient at initialization. We offer a potential theoretical explanation for this empirical phenomenon in terms of certain data-dependent hidden equivalences, emphasizing the connection between these equivalences and the geometry of the loss landscape. We also establish a theoretical upper bound on functional dimension in terms of the number of linear regions sampled by the data set.

## 1 INTRODUCTION

The neural tangent kernel (NTK) has emerged as a powerful tool for understanding the training dynamics and generalization properties of neural networks, especially in the infinite width limit Jacot et al. (2018); Lee et al. (2019). The spectrum of the NTK, in particular, has been shown to play a key role, and significant theoretical progress has been made in obtaining closed-form expressions for this spectrum Murray et al. (2023); Nguyen & Mondelli (2020); Nguyen et al. (2021). A full-rank NTK ensures a well-conditioned optimization problem, leading to efficient training and convergence Arora et al. (2019); Allen-Zhu et al. (2019). However, we still do not understand the effects of finite-width corrections. Indeed, NTK theory has fallen short in predicting how real-world neural networks evolve when training on concrete data sets Geiger et al. (2019); Lee et al. (2020), and the Gram matrix of the NTK - referred to as the *empirical NTK* in the literature - frequently evolves significantly during training in a way that differs from the infinite-width predictions. Our goals here are:

(1) Track the evolution of the empirical NTK during training on synthetic and real-world data sets of increasing complexity;

(2) Relate this evolution to the task complexity; and

(3) Relate the empirical NTK to a complementary theoretical framework involving *data-dependent parameter space symmetries* and their impact on the optimization dynamics of neural networks.

The starting point of our investigation are the following observations:

(1) The *rank* of the empirical NTK on a fixed batch of data agrees with the batch *functional dimension* (cf. Grigsby et al. (2022)), which can be viewed informally as the *effective local parametric dimension* on the batch;

(2) It has been observed empirically that for networks that are deeper than they are wide, the batch functional dimension is much lower at initialization than predicted by the existing theory of parameter space symmetries Grigsby et al. (2023).

Empirically we find:

- *For deep ReLU networks, the empirical NTK has low rank at initialization, rises steadily during the early epochs of training, and then plateaus or gradually decreases.* The rank of the empirical NTK is precisely the batch functional dimension on the data set. In experiments the batch functional dimension at initialization is consistently much smaller than the number of data points for over-parameterized deep ReLU networks. See Figures 2 and 9.

- *The rank of the empirical NTK tracks data set complexity during training.* Training with a small positive weight decay tends to select a function whose complexity, as measured by functional dimension, reflects the complexity of the data set. More complex data sets tend to result in trained functions that have higher functional dimension. See Figures 2 and 3.

- *Weight decay has a damping effect on the rank of the empirical NTK.* That is, higher (constant) weight decay leads to trained functions with lower functional dimension. See Figure 4).

- *Number of linear regions sampled by the dataset is strongly correlated with the rank of the empirical NTK.* See Figures 5 and 6.

We suggest two theoretical mechanisms encouraging a rank-deficient NTK, and relate these mechanisms to the existence of hidden parameter space symmetries:

- *Hidden data-dependent parameter space symmetries encourage a rank-deficient empirical NTK.* Restricted activation patterns cause ReLU networks to behave like smaller networks with more parameter space symmetries. When hidden neurons in a network are either always-active or always-inactive on a batch of data, then the network behaves like a mixed linear-ReLU subnetwork, which enlarges the dimension of the space of symmetries to be *quadratic* rather than *linear* in the number of hidden neurons (Proposition A.10).

- *Fewer linear regions encourage a rank-deficient empirical NTK.* We prove a theoretical upper bound (Proposition **??**) on the batch functional dimension in terms of the number of linear regions sampled by the data set. In the case of architectures with input dimension 1, as in the case of our univariate experiments, the bound asserts that the functional dimension is bounded above by twice the number of linear regions sampled by the batch. Attaining this upper bound would require that the data set and parameter satisfy very specific constraints, so it is unlikely for a typical parameter to attain the bound. In our experiments the number of intervals sampled is typically greater than the batch functional dimension.

## 2 RELATED WORK

**Neural Tangent Kernel (NTK) training and generalization at infinite and finite width:** The NTK was first defined and studied in Jacot et al. (2018), where it was established that in the infinite width limit gradient flow is entirely determined by the NTK and can be described via kernel gradient flow. See also Lee et al. (2019). In Canatar et al. (2021) and Liang & Rakhlin (2020), the authors establish generalization bounds for trained infinitely wide networks from measures that track how rapidly the NTK spectrum decays. These results are only for networks in the infinite width limit, since they assume that the NTK does not evolve during training. The questions and approach we take is closest in philosophy to Baratin et al. (2021), which provides empirical and theoretical evidence that neural feature alignment as measured by the *effective rank* - a continuous version of matrix rank - of the empirical NTK is responsible for favorable generalization behavior. In the very recent paper Grigsby & Lindsey (2024), the authors conjecture that functional dimension (the maximal rank of the empirical NTK) for finite width networks is equal to a local complexity measure called the persistent pseudodimension, from which generalization bounds should be extractable.

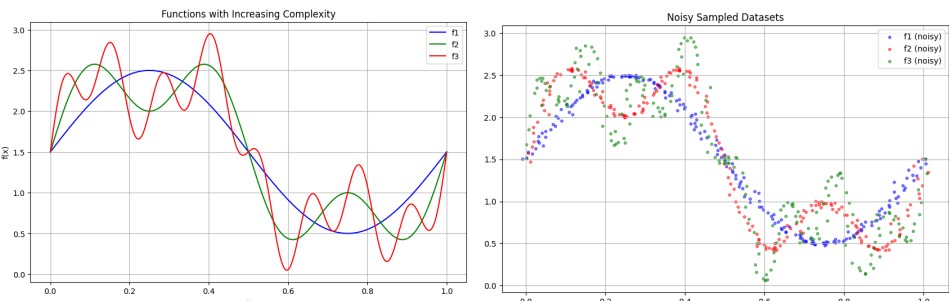

Figure 1: Graphs of three univariate functions of increasing complexity (left); datasets obtained by uniformly sampling 100 points for each function, and then adding Gaussian noise with std=0.01 to both inputs and outputs (right).

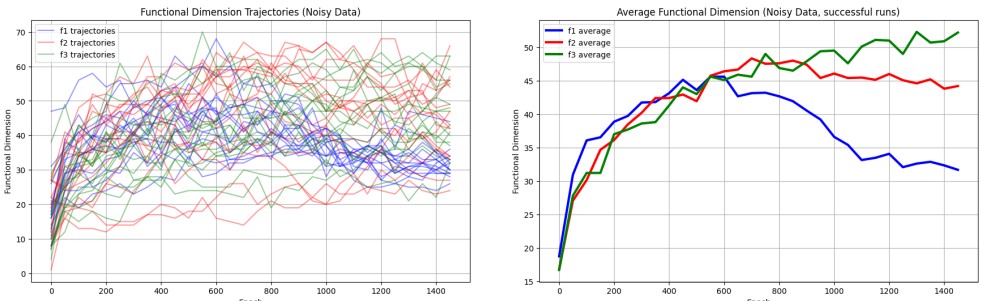

Figure 2: Functional dimension tracks task complexity on the synthetic data. Trajectories tracking functional dimension for 15 randomly initialized training runs per dataset, with functional dimension computed every 50 epochs (left). Averages over the successful ($r^2$ score $> 0.9$) trajectories (right).

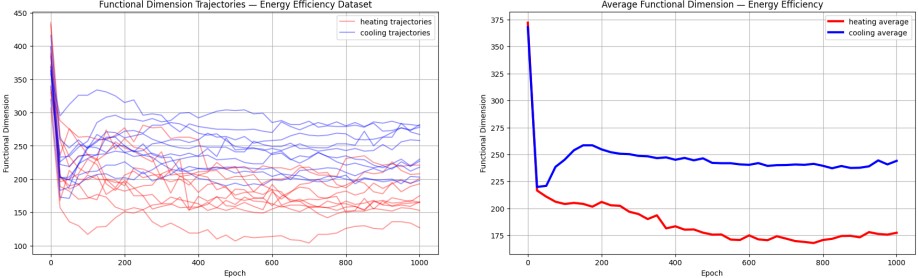

Figure 3: Functional dimension tracks task complexity on the UCI Energy Efficiency dataset: cooling load prediction is widely regarded to be more difficult than heating load prediction (see Section 5). Trajectories tracking functional dimension over 10 randomly initialized training runs (each with a shared random seed for heating vs. cooling), with functional dimension computed every 25 epochs (left); average trajectories (right).

In Hanin & Nica (2020), the authors study the NTK at finite width and depth, arguing that sufficiently deep wide networks can learn data-dependent features even in the so-called "lazy training" regime associated to very wide networks at fixed depth. In Huang & Yau (2020), the authors define and study the dynamics of gradient descent for finite-width networks under a so-called neural tangent hierarchy of differential equations, for which the NTK gives an approximation.

**NTK eigenfunctions and spectrum analysis:** In Xie et al. (2017), the authors study the dyanamics of the training loss for shallow ReLU neural networks, establishing a connection between the minimum singular value of the empirical NTK and the decay rates of both the training loss and the kernel spectrum associated to the arc-cosine kernel defined by Cho & Saul (2009). In Nguyen & Mondelli (2020) and Nguyen et al. (2021) the authors perform a spectrum analysis for deep ReLU networks with certain architecture restrictions, and in Murray et al. (2023), the authors derive a power series

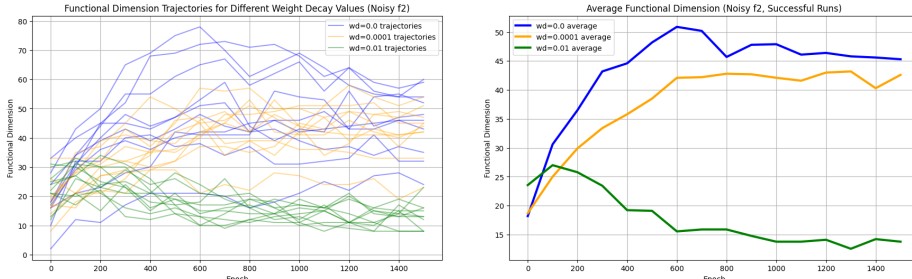

Figure 4: Weight decay suppresses functional dimension. We tracked functional dimension for the synthetic $f2$ data set during 10 randomly initialized training runs for each of 3 weight decay rates: 0, 1e-4, and 1e-3. Right: averages over the successful trajectories (all except one for weight decay 1e-4).

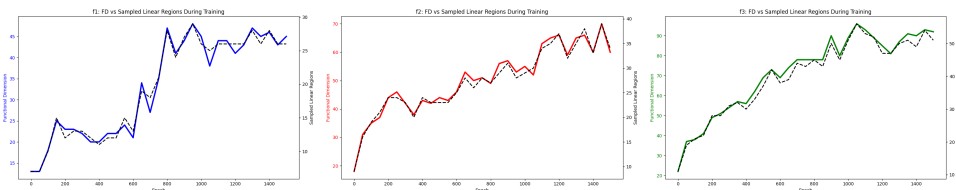

Figure 5: Functional dimension is closely correlated with the number of linear regions sampled. These plots show a single training run for each synthetic dataset, with functional dimension (solid blue/red/green) and number linear regions (dashed black) computed every 50 epochs.

expansion for the NTK of arbitrarily deep feedforward networks in the infinite width limit that allows them to extract the eigenvalue spectrum.

**Parameter space symmetries and optimization:** Two largely-independent approaches to studying the relationships among parameter space symmetries, the geometry of the loss landscape, and the so-called *neuromanifold* (true function space after quotienting by symmetries) have emerged, as described in the recent survey papers Zhao et al. (2025), Marchetti et al. (2025), and the many references therein. The approach we take here is more closely aligned with the first survey article, although we are interested in connections to the second. We are not aware of any prior work explicitly discussing a relationship between the NTK spectrum and parameter space symmetries.

## 3 BACKGROUND AND NOTATION

### 3.1 FULLY-CONNECTED FEEDFORWARD RELU NETWORKS

We focus on fully connected neural networks with ReLU activation, denoting by $(n_0, \ldots, n_{d-1}|n_d)$ the architecture with input width $n_0$, hidden layer widths $n_1, \ldots, n_{d-1}$, and output width $n_d$.

Formally, let $\sigma : \mathbb{R}^n \to \mathbb{R}^n$ denote the function that applies the activation function $\mathrm{ReLU}(x) := \max\{0, x\}$ component-wise. For an architecture $(n_0, \ldots, n_{d-1}|n_d)$, we denote the parameter space $\Omega := \mathbb{R}^D$ where a parameter $\theta := (W^1, b^1, \ldots, W^d, b^d) \in \Omega$ consists of the entries of weight matrices $W^\ell$ and bias vectors $b^\ell$ for $\ell = 1, \ldots, d$. Accordingly, $D := \sum_{\ell=1}^d n_\ell(n_{\ell-1} + 1)$. From a parameter $\theta$ we define a neural network function $F_\theta := F^d \circ \ldots \circ F^1$ with layer maps given by:

$$F^\ell(x) := \begin{cases} \sigma(W^\ell x + b^\ell) & \text{for } 1 \le \ell < d \\ W^\ell x + b^\ell & \text{for } \ell = d. \end{cases} \tag{1}$$

To compactify notation, following Masden (2022) we let $F_{(\ell)} := F^\ell \circ \ldots \circ F^1$. We refer to the components of $F_{(\ell)}$ as the *neurons* in the $\ell$th layer. The *pre-activation* map $z_{\ell,i} : \mathbb{R}^{n_0} \to \mathbb{R}$ associated to the $i$th neuron in the $\ell$th layer is given by:

$$z_{\ell,i}(x) = \pi_i \left( W^\ell(F_{(\ell-1)}(x)) + b^\ell \right), \tag{2}$$

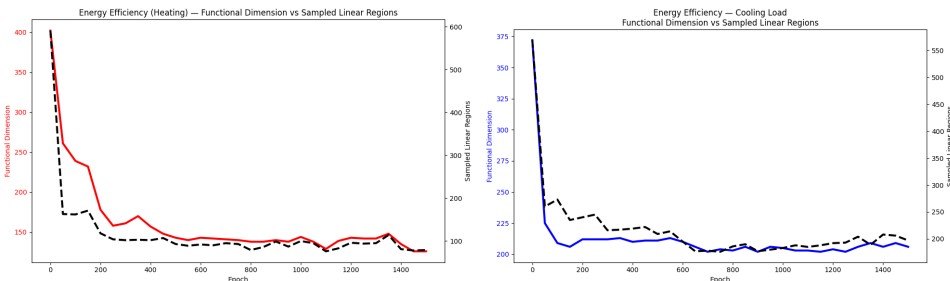

Figure 6: Functional dimension is closely correlated with the number of linear regions sampled for the energy efficiency dataset. A single training run tracking functional dimension for heating/cooling tasks (red/blue) vs number linear regions (dashed black), computed every 50 epochs.

where $\pi_i : \mathbb{R}^{n_\ell} \to \mathbb{R}$ denotes the projection onto the $i$th component.

Given a point $x \in \mathbb{R}^{n_0}$ in the input space we can record its activation status with respect to all $N = \sum_{i=1}^{d-1} n_i$ hidden neurons by computing the $N$–tuple $s(x) = \{-1, 0, +1\}^N$ of pre-activation signs for neurons in the network. Explicitly, the component of $s(x)$ corresponding to the $i$th neuron in the $\ell$th layer is: $s_{\ell,i}(x) := \mathrm{sgn}(z_{\ell,i}(x))$, where $\mathrm{sgn}(z) = \begin{cases} \frac{z}{|z|} & \text{if } z \neq 0 \\ 0 & \text{if } z = 0. \end{cases}$

In the present work, we will also be interested in the activation pattern of each *neuron* in the network with respect to a finite data set $X = \{x_1, \ldots, x_m\}$. For neuron $i$ in layer $\ell$ this is the $m$–tuple $(s_{\ell,i}(x_1), \ldots, s_{\ell,i}(x_m))$. If $s_{\ell,i}(x_j) = +1$ (resp., $= -1$) for all $x_j \in X$ we say that neuron $i$ in layer $\ell$ is *always-active* (resp., *always-inactive*) on the data set $X$.

Our main result (Theorem 1) requires an architecture restriction in order to rule out undesirable behavior. Accordingly, we define:

**Assumption 3.1.** *We say that an architecture $(n_0, \ldots, n_{d-1} \,|\, n_d)$ is* roughly monotonic *if:*

*(i) For all $\ell \in \{1, \ldots, d\}$, we have $n_\ell - 1 \leq n_{\ell-1}$, or*

*(ii) For all $\ell \in \{1, \ldots, d\}$, we have $n_{\ell-1} - 1 \leq n_\ell$.*

In other words, the dimensions of the layers either grow or shrink (up to a linear error) through the network. See Appendix A for more details. An example of an architecture that is roughly monotonic is $(1, 5, 4, 3, 2|1)$ and an example of an architecture that is not roughly monotonic is $(1, 5, 5, 5, 5|1)$.

### 3.2 Spectrum of the NTK for ReLU networks

Recall that a *kernel* $k : \mathbb{R}^{n_0} \times \mathbb{R}^{n_0} \to \mathbb{R}_{\geq 0}$ is a symmetric, positive semi-definite similarity measure on the input space of a function class, most naturally obtained by pulling back an inner product from a *kernel feature map* $\Phi : \mathbb{R}^{n_0} \to \mathcal{H}$ into a Hilbert space $\mathcal{H}$: $k_\Phi(x, y) := \langle \Phi(x), \Phi(y) \rangle$.

In the case of the neural tangent kernel (NTK) associated to a parameter $\theta \in \Omega$ at initialization, $\mathcal{H}$ is the tangent space $T_\theta(\Omega) \cong \mathbb{R}^D$ at that parameter, equipped with its standard inner product, and the feature map $\Phi : \mathbb{R}^{n_0} \to \mathcal{H}$ is the assignment of the parameter gradient vector $\nabla E_z|_\theta$ of the evaluation map at each input vector $z \in \mathbb{R}^{n_0}$. Mercer's Theorem (cf. Schölkopf & Smola (2002)) associates to any kernel $k$ on a compact set $\chi \subseteq \mathbb{R}^{n_0}$ a natural positive semi-definite integral operator $T_k$ on $L_2(\chi)$, defined by $T_k f(\cdot) := \int_\chi k(\cdot, y) f(y) dy$, whose associated eigenfunctions can be viewed as a preferred orthonormal basis of $L_2(\chi)$ associated to the kernel. The relationship between the eigenbasis and spectrum of the NTK operator, optimization dynamics, and generalization has been widely studied, cf. Murray et al. (2023) and the references therein. It is frequently assumed that at initialization the empirical NTK will have full rank–i.e., will be equal to the number of data points in the overparameterized setting.

### 3.3 Batch functional dimension and the empirical NTK for ReLU networks

The (batch) functional dimension of a parameter $\theta \in \Omega$ on a finite data set $Z \subseteq \mathbb{R}^{n_0}$ was defined (away from a Lebesgue measure $0$ set) for ReLU neural network classes in Grigsby et al. (2022), see also Stock (2023); Bona-Pellissier et al. (2022; 2024). It is the rank of the Jacobian matrix with respect to the parameters of the evaluation map on the batch $Z$: $\mathrm{rk}(\mathbf{J}E_Z|_\theta)$. When the output dimension is 1, $Z = \{z_1, \ldots, z_m\}$, and the parametric dimension is $D$, $\mathbf{J}E_Z$ is an $m \times D$ matrix whose rows are $\nabla E_{z_i}|_\theta$, the neural tangent kernel feature maps at the $m$ points of $Z$.

In Section 6 of Grigsby et al. (2022) it is noted that the Gram matrix of the NTK at $\theta$ on a batch $Z$ is $(\mathbf{J}E_Z)(\mathbf{J}E_Z)^T$. Moreover, it is a well-known linear algebra fact that for all matrices $M$ over $\mathbb{R}$:

$$\mathrm{rk}(M) = \mathrm{rk}(MM^T),$$

so the rank of the Gram matrix, $(\mathbf{J}E_Z)(\mathbf{J}E_Z)^T$, of the NTK at $\theta$ on a batch $Z$ is precisely the batch functional dimension of $\theta$ on the batch $Z$.

## 4 Parameter space symmetries and functional dimension

One take-away from the experiments described here is that functional dimension at initialization is often far below what theory predicts. In this section, we (i) recall the previously-established connection between functional dimension and *parameter space symmetries*, (ii) propose activation sparsity as a mechanism for explaining the gap between theory and experiment, and (iii) provide novel data-dependent bounds (Theorem 1 and Proposition 4.6) on functional dimension (aka the rank of the empirical NTK) in this setting. We do not claim that these theoretical results provide a complete explanation for the empirical phenomena we observe. We merely describe succinctly existing theoretical frameworks and show how they can be used to give novel bounds that may partially explain the rank gap.

It is well-known that function representation in neural architectures is highly redundant; many parameter settings give rise to the same function. This redundancy is governed by the architecture's *parameter space symmetries*, Godfrey et al. (2022); Zhao et al.. In Grigsby et al. (2022; 2023); Grigsby & Lindsey (2024) it is noted that in favorable situations the local dimension of parameter space symmetries is complementary to functional dimension. Explicitly (but informally), for a parameter $\theta$: if $D$ is the total parametric dimension, $d_\theta$ is the functional dimension at $\theta$ and $s_\theta$ is the dimension of the space of symmetries near $\theta$, then $D = d_\theta + s_\theta$. The picture is that parameter space is decomposed locally into directions that can change the function (hence contribute to functional dimension) and directions that preserve the function (hence contribute to the local space of symmetries).

Following Serra et al. (2020), we call two neural network functions[1] $F_i : \mathbb{R}^{n_0} \to \mathbb{R}^{n_d}$ for $i = 1, 2$ *equivalent* if $F_1(x) = F_2(x)$ for all $x \in \mathbb{R}^{n_0}$. If $X \subsetneq \mathbb{R}^{n_0}$ is a proper subset, we say $F_i$ are equivalent for $i = 1, 2$ *relative to $X$* if $F_1(x) = F_2(x)$ for all $x \in X$. Informally, a *global parameter space symmetry* is a map from parameter space to itself that sends all functions to equivalent functions. A *data-dependent parameter space symmetry* is a map that induces equivalences only relative to a proper subset $X$. In the remainder of this section, we formalize these concepts and show how they can be used to give a potential theoretical explanation for our empirical observations.

### 4.1 Group Actions, Orbits, and Hidden Equivalences

Feedforward ReLU architectures are well-known to be invariant under *positive scaling* and *permutation* of hidden neurons (cf. (Rolnick & Kording, 2020; Bui Thi Mai & Lampert, 2020; Grigsby et al., 2023)). Letting $N = \sum_{\ell=1}^{d-1} n_\ell$ be the number of hidden neurons in a ReLU network, this typically results in *at least* an $N$–dimensional space of parameter choices for each function representable by the class. The above well-known fact is an incarnation of a much more general phenomenon that is central to establishing upper bounds on the rank of the empirical NTK.

It is profitable to view many redundancies as arising directly from the computational structure of the function class. Following Zhao et al., we note that the parameter space of a feedforward network

---

[1] We do *not* assume that the functions are associated to networks of the same architecture, but they necessarily have the same input and output dimensions.

architecture $(n_0, \ldots, n_{d-1}|n_d)$ (for any choice of activation function) admits a natural action of the *hidden symmetry group*, $G_{hid} := GL_{n_1} \times \ldots \times GL_{n_{d-1}}$. Here, $GL_n$ denotes the general linear group of invertible $n \times n$ matrices over $\mathbb{R}$, so any $g = (g_1, \ldots, g_{d-1}) \in G_{hid}$ gives rise to a map $g \cdot - : \Omega \to \Omega$ defined as follows. If $\theta = (W^1, b^1, \ldots, W^d, b^d) \in \Omega$, then:

$$g \cdot (W^\ell, b^\ell) := (g_\ell W^\ell g_{\ell-1}^{-1}, g_\ell b^\ell), \tag{3}$$

where we set $g_0 := \mathrm{Id}_{n_0}, g_d := \mathrm{Id}_{n_d}$, the identity matrices of the appropriate dimensions.

One can and should understand this particular action as arising more naturally by performing a conjugation (change-of-basis) operation on the $\ell$th hidden layer and pulling the conjugating matrices past the activation layers so that they are now being multiplied by the *parameters* rather than the hidden vectors themselves. The observant reader will recognize this action as inducing equivalences for deep *linear* networks, since the overall function will be invariant under this action as long as there is no activation function. If an activation function is present, we simply restrict attention to the largest subgroup of $G_{hid}$ that commutes with the component-wise application of the activation function, and we arrive at the same conclusion. Accordingly:

**Definition 4.1.** *We define the* hidden equivalence group, $H_{eq} \subseteq G_{hid}$ *for a fixed architecture and batch* $X \subseteq \mathbb{R}^{n_0}$ *of input data to be the largest subgroup of the hidden symmetry group that commutes with the component-wise application of the activation function for all* $x \in X$.

In this case, each $H_{eq}$–orbit, $H_{eq}\theta := \{h \cdot \theta \mid h \in H_{eq}\}$, consists of equivalent parameters; see Zhao et al. and Section A in the Appendix for more details.

$G_{hid}$ and $H_{eq}$ are examples of *Lie groups*, and the assignment of a smooth map on a vector space associated to every element of a Lie group is an example of a *Lie group action*. Lie groups - smooth manifolds with a group structure that interacts cleanly with the smooth structure - are familiar objects to differential geometers. They have many useful properties that are by now standard in the mathematics literature but which would be difficult to establish directly in the absence of this history.

In particular, any time a Lie group $H$ acts on a smooth manifold (in this case, on the parameter space of a neural architecture), the space decomposes into *orbits* under the action, and the dimensions of these orbits can vary according to local properties. The powerful classical *orbit-stabilizer theorem* for Lie group actions (Corollary A.6, cf. Lee (2013)) tells us that $H\theta$ is diffeomorphic to the quotient space, $H/\mathrm{Stab}_H(\theta)$, where

$$\mathrm{Stab}_H(\theta) := \{h \in H \mid h \cdot \theta = \theta\}$$

is the *stabilizer* of $\theta$. In particular, the dimension of the orbit $H\theta$ is the dimension of the Lie group $H$ minus the dimension of the stabilizer of any parameter $\theta$ in the orbit:

$$\dim(H\theta) = \dim(H) - \dim(\mathrm{Stab}_H(\theta)). \tag{4}$$

It is immediate that all functions in any $H_{eq}$-orbit of any parameter $\theta \in \Omega$ are equivalent on $X$. In particular, as long as the stabilizer of a parameter is trivial (i.e., consists only of the identity element of the Lie group), the dimension of $H_{eq}$ will give us a lower bound on the dimension of the group of parameter space symmetries. It follows that if the hidden equivalence group is larger than expected, then level sets of the loss function will also be larger than expected (in the trivial stabilizer case), since equivalent functions yield the same empirical loss for any loss function:

**Lemma 4.2.** *Let* $\mathcal{C} \subseteq \Omega$ *be the critical locus for any empirical loss function* $\mathcal{L}_X : \Omega \to \mathbb{R}$. *If there exists* $\theta \in \mathcal{C}$ *with trivial stabilizer (for which* $\mathrm{Stab}_{H_{eq}}(\theta) = \{Id\}$), *then* $dim(\mathcal{C}) \geq dim(H_{eq})$.

*Remark* 4.3. The *expected* dimension of the hidden equivalence group for a ReLU network of architecture $(n_0, n_1, \ldots, n_{d-1}|n_d)$ is $N = \sum_{\ell=1}^{d-1} n_\ell$, the number of hidden neurons. Indeed, letting $PD_+(n)$ denote the group of $n \times n$ matrices representable as a product $PD$, where $P$ is an $n \times n$ permutation matrix and $D$ is an $n \times n$ diagonal matrix with strictly positive entries on the diagonal, the subgroup $PD_+(n_1) \times \ldots \times PD_+(n_{d-1})$ of the hidden symmetry group commutes with the component-wise application of ReLU (Zhao et al.; Godfrey et al., 2022), resulting in the familiar *positive scaling invariance* and *permutation invariance* for ReLU networks mentioned above.

Lemma 4.2, which follows from the orbit-stabilizer theorem and the results in Appendix A, tells us that if the hidden equivalence group *on a batch of data $X$* includes a subgroup of dimension larger than $N$ and the stabilizer is known to be trivial in general, then the symmetry group will be larger than the expected dimension $N$, decreasing the functional dimension on the batch. In the following section, we describe one way this can occur.

## 4.2 RESTRICTED ACTIVATION PATTERNS INDUCE HIDDEN EQUIVALENCES

If there are neurons for $F_\theta$ that are either always-active or always-inactive on a data set $X \subseteq \mathbb{R}^{n_0}$, then ReLU acts locally as the Identity function at always-active neurons and effectively ignores always-inactive neurons. This observation motivates the following, cf. Serra et al. (2020):

**Definition 4.4.** *Let $\sigma_{[1:k]} : \mathbb{R}^n \to \mathbb{R}^n$ denote the function that applies the activation function ReLU(x) (resp., $Id(x)$) to the first $k$ components (resp., to the last $n-k$ components). A mixed ReLU-linear neural network of architecture $(n_0, (n_1^R, n_1^L), \ldots, (n_{d-1}^R, n_{d-1}^L)|n_d)$ is a neural network of architecture $(n_0, n_1^R + n_1^I, \ldots, n_{d-1}^R + n_1^L \mid n_d)$ whose $\ell$th layer map is $F^\ell(x) := \sigma_{[1:n_\ell^R]}(W^\ell x + b^\ell)$.*

The following results, whose formal statements and proofs appear in Appendix 4.1, together tell us that the dimension of the hidden equivalence group is *quadratic*, rather than *linear* in the number of neurons with fixed activation status with respect to the data set $X$.

**Proposition 4.5.** *The hidden equivalence group of a mixed ReLU-linear neural network of architecture $(n_0, n_1^R + n_1^I, \ldots, n_{d-1}^R + n_{d-1}^L \mid n_d)$ has dimension at least $d = \sum_{\ell=0}^{d-1} n_\ell^R + \sum_{\ell=1}^{d-1} (n_\ell^L)^2$.*

**Theorem 1.** *Let $\theta \in \Omega$ be almost any parameter[2] in a roughly monotonic feedforward ReLU network architecture, $(n_0, \ldots, n_{d-1}|n_d)$, with parametric dimension $D$, and let $X \subseteq \mathbb{R}^{n_0}$ be a subset of the domain for which $n_\ell^{fixed}$ of the hidden neurons from layer $\ell$ are either always-active or always-inactive on all of $X$. Then the batch functional dimension of $\theta$ on $X$ is at most $D - d$, where*

$$d = \sum_{\ell=1}^{d-1} \left( n_\ell - n_\ell^{fixed} \right) + \sum_{\ell=1}^{d-1} \left( n_\ell^{fixed} \right)^2.$$

The idea of the proof of Theorem 1 is that near $\theta$, the space of data-dependent parameter space symmetries on $X$ looks locally like the hidden equivalence group of a mixed ReLU-linear network with $n_\ell^R = n - n_\ell^{fixed}$ and $n_\ell^L = n_\ell^{fixed}$. The *roughly monotonic* condition on the architecture insures that the stabilizer of the action is trivial, so the dimension of the orbit is the dimension of this group.

## 4.3 NUMBER OF LINEAR REGIONS SAMPLED CONSTRAINS BATCH FUNCTIONAL DIMENSION

The number of linear regions of a piecewise-linear function $f_\theta$ sampled by a batch of inputs $Z$ is the number of distinct affine pieces of $f_\theta$ that are actually encountered on the set $Z$.

**Proposition 4.6.** *For any batch of inputs $Z$, for a full measure set of parameters $\theta$, the batch functional dimension at $\theta$ is at most (input dimension + 1) times the number of linear regions of $f_\theta$ sampled by $Z$.*

The proof of Proposition 4.6 is in Appendix C.

## 5 EXPERIMENTAL SETUP

**Functions and data sets:**

*Synthetic univariate data with noise:* We use univariate (input dimension 1) functions so we can easily plot their evolution during training.

The univariate functions $f_1, f_2, f_3 : [0, 1] \to \mathbb{R}$ are defined as follows:

$$f_1(x) = 1.5 + \sin(2\pi x)$$
$$f_2(x) = 1.5 + \sin(2\pi x) + 0.5 \sin(4\pi x)$$
$$f_3(x) = 1.5 + \sin(2\pi x) + 0.5 \sin(4\pi x) + 0.4 \sin(16\pi x).$$

For each function $f_i$, we create a data set $\mathcal{D}_i = \{x_j, y_j\}_{j=1}^{100} \subset \mathbb{R} \times \mathbb{R}$ by sampling the domain at 100 equally spaced points, and then adding Gaussian noise (with std= 0.01, mean= 0) to both the inputs and outputs. The purpose of the vertical shift $1.5$ in the functions is to make all functions strictly positive. Figure 1 shows the graphs and datasets for the functions $f_i$.

---

[2]$\theta$ is in the complement of a Lebesgue measure 0 set defined in the proof of Proposition A.8 in the Appendix.

We also tracked functional dimension for bivariate synthetic data sets of increasing complexity. See Appendix B.1.

*UCI Energy Efficiency data:* In addition to the synthetic datasets above, we include a real-world benchmark drawn from the UCI Energy Efficiency Dataset (Tsanas & Xifara, 2012a;b). This dataset consists of 768 building-design observations, each described by eight geometric and thermophysical features (relative compactness, surface area, wall area, roof area, overall height, orientation, glazing area, and glazing-area distribution). The dataset provides two regression targets, the heating and cooling load requirements associated with each design.

A consistent finding in the literature is that Heating Load (HL) is easier to model than Cooling Load (CL). In the original study associated with this dataset (Tsanas & Xifara, 2012a;b), the authors report systematically smaller prediction errors for HL across multiple regression techniques, including linear models and nonlinear ensemble methods. Later surveys of data-driven building-load prediction document the same pattern across a broad range of linear and nonlinear ML algorithms, cf. Figure 9 in Salami et al. (2023). Heating Load is uniformly approximated more accurately, and the accuracy gap between linear and nonlinear models is wider for CL than for HL.

We chose to use the energy efficiency dataset because it is small enough that full-batch functional dimension can be computed exactly, includes enough diverse input configurations to meaningfully probe the network's behavior across the input space, and offers two regression tasks.

**Training setup:** For all experiments, we used the fixed architecture (input dimension, $10, 10, 10, 10, 1$) of fully-connected, feedforward ReLU neural networks. For the univariate experiments, the associated number of trainable parameters is $D = 361$, for the bivariate experiments, the parametric dimension is $D = 371$, and for the Energy Efficiency data set $D = 431$. We compute functional dimension periodically throughout the training as indicated in the figure captions. We train all networks networks using the Adam optimizer with MSELoss, learning rate 0.01, weight decay 1e-4, and shuffled minibatches of batchsize 16 (unless specified otherwise). We initialize each training run using a variant of the He (also called Kaiming) initialization – all weights and all biases for a given layer are selected randomly from a normal distribution with mean 0 and standard deviation $\sqrt{\frac{2}{\text{fan\_in}}}$, where fan\_in is the number of input features for that layer.

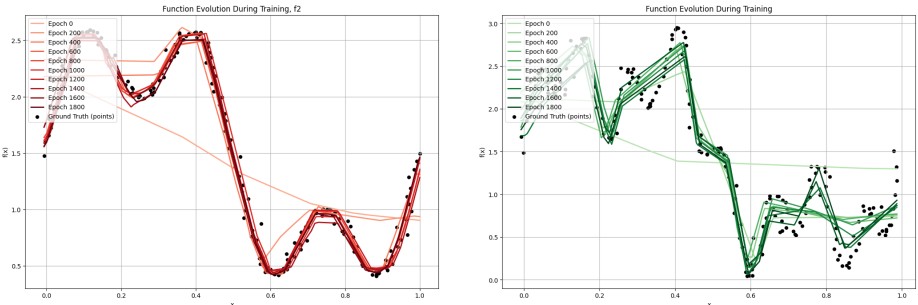

Figure 7: Function evolution for a single training run, for functions $f_2$ and $f_3$. Although the networks are overparameterized (100 data points and 361 parameters), the learned functions do not overfit.

**Batch functional dimension computation:** At specified epochs during training, we compute the batch functional dimension using the inputs for the entire data set $\mathcal{D}_i$ as the batch. To do this, we first compute the matrix

$$\mathbf{J}E_{\mathcal{D}_i}(\theta) := \begin{bmatrix} \nabla_\theta f(x_1; \theta) \\ \nabla_\theta f(x_2; \theta) \\ \vdots \\ \nabla_\theta f(x_m; \theta) \end{bmatrix}$$

where each $x_j$ is one of the inputs of a point $(x_j, y_j) \in \mathcal{D}_i$ and $f(\cdot, \theta)$ is the function determined by the parameter $\theta$. We then compute the batch functional dimension as

$$\dim_{\text{ba.fun}}(\theta) = \text{torch.linalg.matrix\_rank}(\mathbf{J}E_{\mathcal{D}_i}(\theta)).$$

Note that the command torch.linalg.matrix_rank computes the number of "non-zero" singular values, where a singular value is considered non-zero if it is greater than the default error tolerance for floating point calculations. This default threshold is defined for a $n \times m$ matrix $A$ as

$$\text{threshold}(A) \coloneqq \max(n, m) \cdot \sigma_{max} \cdot \epsilon$$

where $\sigma_{max}$ is the largest singular value of $A$ and $\epsilon$ represents machine precision, which is roughly $1.19 * 10^{-7}$ for dtype float32. Thus, our computation of batch functional dimension will set to 0 any singular values sufficiently small with respect to the maximum possible rank of the Jacobian matrix (100 for our 1D experiments and 400 for our 2D experiments), and the largest singular value, which empirically were $\approx 10^3$ for our 1D experiments and $\approx 10^4$ for our 2D experiments. The calculation of the rank is therefore setting to 0 singular values below (roughly) 0.01 for our 1D experiments and 0.1 for our 2D experiments (both typically $10^{-5}$ times $\sigma_{max}$).

**Number of linear regions sampled:** The number of linear regions sampled by the dataset is computed by counting the distinct ReLU activation patterns across all inputs: for each sample, we record a binary vector indicating which neurons in each layer are active, and each unique pattern corresponds to a unique linear region of the piecewise-linear function implemented by the network.

## 6 EXPERIMENTAL RESULTS

**The empirical NTK is typically rank deficient at initialization.** This observation is supported by Figures 2, 3, and 9. For the univariate experiments (Figure 2), there are 100 data points and the parametric dimension is $D = 361$, so the functional dimension is the rank of a $100 \times 361$ matrix, yet at initialization the functional dimension is on average below 20. Similarly, for the Energy Efficiency experiments (Figure 3), the functional dimension is the rank of a $768 \times 431$ matrix, while the functional dimension at initialization is on average below 400. For the bivariate experiments (Figure 9), the functional dimension at initialization is on average less than 60 for a $100 \times 371$ matrix.

**Weight decay suppresses batch functional dimension of trained networks.** See Figure 4.

**Evolution of functional dimension during training.** In all experiments, the functional dimension begins well below the number of data points and rises steadily during early training. In later epochs, however, the trend changes: the growth rate diminishes, and the functional dimension frequently decreases gradually as weight decay exerts a regularizing influence.

**Batch functional dimension tracks dataset complexity.** Figures 2 supports the conclusion that for the univariate datasets, on average, training tends to select a function whose complexity, as measured by functional dimension, reflects the complexity of the data set. That is, the rank of the empirical NTK correlates positively at the end of training with the complexity of the function the model is learning. Figures 3 and 9 demonstrate the same trend for the Energy Efficiency data set and the synthetic bivariate data set.

**Batch functional dimension is correlated with number of regions sampled.** Proposition **??** proves that, except for a measure 0 situation, the functional dimension is bounded by the number of linear regions sampled by the dataset times $(n_0 + 1)$, where $n_0$ is the input dimension. This result is confirmed empirically in Figures 5 and 6.

## 7 CONCLUSIONS

We have performed both a theoretical and empirical investigation of the behavior of the empirical NTK during training of deep ReLU networks on synthetic and real-world datasets, assembling empirical evidence that the rank of the empirical NTK in this setting is (i) lower-than-expected at initialization, and (ii) tracks task complexity. We provide a possible theoretical explanation for this phenomenon by relating the rank of the empirical NTK to the functional dimension, whose behavior for ReLU networks has been related to a growing body of literature on parameter space symmetries.

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

## A  LIE GROUP ACTIONS, HOMOGENEOUS SPACES, ORBITS, AND STABILIZERS

We recall some classical results about Lie group actions on smooth manifolds, following the treatment in Lee (2013).

**Definition A.1.** *A* Lie group *is a smooth manifold $G$ that is also a group, for which the group operations are all smooth maps. That is, $G$ is endowed with a multiplication map*

$$m : G \times G \to G \qquad\qquad m(g, h) = gh$$

*and an inversion map*

$$i : G \to G \qquad\qquad i(g) = g^{-1},$$

*and both $m$ and $i$ are smooth (derivatives of all orders are well-defined).*

**Definition A.2.** *If $G$ is a Lie group with identity element Id, and $\Omega$ is a smooth manifold, a smooth left action of $G$ on $\Omega$ is a map*

$$\psi : G \times \Omega \to \Omega \qquad\qquad (g, \theta) \mapsto g \cdot \theta$$

*satisfying:*

- *$\psi$ is a smooth map.*

- *$g_1 \cdot (g_2 \cdot \theta) = (g_1 g_2) \cdot \theta$ for all $g_1, g_2 \in G, \theta \in \Omega$,*

- *Id $\cdot \theta = \theta$ for all $\theta \in \Omega$.*

**Definition A.3.** *Let $G$ be a Lie group acting smoothly on a smooth manifold $\Omega$.*

- *The action of $G$ on $\Omega$ is said to be* transitive *if for all $\theta, \theta' \in \Omega$ there exists $g \in G$ such that $g \cdot \theta = \theta'$.*

- *For $\theta \in \Omega$, the* orbit *of $\theta$ under the action of $G$ is the set of points in $\Omega$ obtainable by applying an element $g \in G$ to $\theta$:*

$$G\theta := \{g \cdot \theta \mid g \in G\}.$$

- *For $\theta \in \Omega$, the* stabilizer *of $\theta$ is the set*

$$Stab_G(\theta) := \{g \in G \mid g \cdot \theta = \theta\}.$$

**Definition A.4.** *A homogeneous $G$–space is a smooth manifold $M$ equipped with a transitive action of a Lie group $G$.*

It is immediate from the definitions that if a Lie group $G$ acts smoothly on a smooth manifold $\Omega$, then for every $\theta \in \Omega$, every $G$–orbit $G\theta$ is a homogeneous space.[3]

Proofs of the following results can be found in the section on *Homogeneous spaces* in Chapter 9 of Lee (2013), which mostly rely on the equivariant rank theorem for Lie groups (Theorem 9.7 in Lee (2013)).

**Lemma A.5.** *(Lemma 9.23 of Lee (2013)) If $G$ is a Lie group acting smoothly on $\Omega$, $Stab_G(\theta)$ is a closed Lie subgroup of $G$ for every $\theta \in \Omega$.*

**Theorem 2.** *(Theorem 9.22 of Lee (2013) Let $G$ be a Lie group, and let $H \subseteq G$ be a closed Lie subgroup of $G$. The left coset space $G/H$ has a unique smooth manifold structure such that the quotient map $\pi : G \to G/H$ is a smooth submersion. Moreover, $G/H$ is also a homogeneous $G$–space with respect to the natural $G$–action on the quotient group.*

**Theorem 3.** *(Theorem 9.24 of Lee (2013) Let $G$ be a Lie group and $M$ a homogeneous $G$–space. Then the coset space (quotient space) $G/Stab_G(\theta)$ is diffeomorphic to $M$.*

**Corollary A.6.** *(Orbit-stabilizer theorem) Let $G$ be a Lie group acting smoothly on a manifold $\Omega$, and let $G\theta$ be the $G$–orbit of $\theta \in \Omega$. Then $G/Stab_G(\theta)$ is diffeomorphic to $G\theta$. In particular,*

$$dim(G\theta) = dim(G) - dim(Stab_G(\theta).$$

The following lemma is immediate from the definitions.

**Lemma A.7.** *Let $X$ be a smooth manifold, $G$ a Lie group acting on $X$, and $H \subseteq G$ a Lie subgroup of $G$. For $\theta \in X$, if $Stab_G(\theta)$ is trivial, then so is $Stab_H(\theta)$.*

*Proof.* $\text{Stab}_H(\theta)$ is by definition a subgroup of $\text{Stab}_G(\theta)$, so if $\text{Stab}_G(\theta)$ is the trivial subgroup, then so is $\text{Stab}_H(\theta)$. $\qquad\square$

We will sometimes need the following architecture restriction in order to deduce properties of the functional dimension from our knowledge of a particular Lie group action:

---

[3]Beware that this does *not* imply that every $G$–orbit is a *smoothly imbedded submanifold* of $\Omega$. See the examples in the section on Proper Actions in Chapter 9 of Lee (2013).

**Assumption A.1.** *We say that an architecture $(n_0, \ldots, n_{d-1} \,|\, n_d)$ is* roughly monotonic *if:*

*(i) For all $\ell \in \{1, \ldots, d\}$, we have $n_\ell - 1 \leq n_{\ell-1}$, or*

*(ii) For all $\ell \in \{1, \ldots, d\}$, we have $n_{\ell-1} - 1 \leq n_\ell$.*

In other words, the dimensions of the layers either grow or shrink (up to a linear error) through the network. In practice, architectures are typically roughly rectangular, so this restriction is mild.

**Proposition A.8.** *Let $\Omega$ be the parameter space for a ReLU neural network of architecture $(n_0, n_1, \ldots, n_{d-1} \,|\, n_d)$. If Assumption A.1 holds, then for almost all $\theta \in \Omega$, $Stab_{G_{hid}}(\theta)$ is trivial.*

In other words, as long as the dimensions of the layers don't grow too fast (condition (i)) or shrink too fast (condition (ii)) as you move through the network, every parameter away from a Lebesgue measure 0 set in parameter space has trivial stabilizer under the action of the hidden symmetry group.

*Proof.* Recall (Equation 3) that

$$g = (g_1, \ldots, g_d) \in G_{hid} := GL_{n_1} \times \ldots \times GL_{n_{d-1}}$$

acts on $\theta = (W^1, b^1, \ldots, W^d, b^d) \in \Omega$ by:

$$(W^\ell, b^\ell) \quad \rightarrow \quad (g_\ell W^\ell g_{\ell-1}^{-1}, g_\ell b^\ell),$$

so if $g$ is in the stabilizer of $\theta$, then (recalling that for convenience we set $g_0 = g_d = \text{Id}$) we have:

$$
\begin{aligned}
g_1 W^1 &= W^1 & g_1 b^1 &= b^1 \\
g_2 W^2 g_1^{-1} &= W^2 & g_2 b^2 &= b^2 \\
&\cdots & &\cdots \\
g_{d-1} W^{d-1} g_{d-2}^{-1} &= W^{d-1} & g_{d-1} b^{d-1} &= b^{d-1} \\
W^d g_{d-1}^{-1} &= W^d &
\end{aligned}
$$

Now note that by the first line of equations above, 1 must be an eigenvalue of $g_1$, and the dimension of the eigenspace of 1 must be at least the the rank of the matrix, $(W^1 \ b^1)$, whose columns are all in the in the eigenspace of 1 for $g_1$. But if we assume that $g_1 \neq \text{Id}$, then by the fact that each square matrix has a unique Jordan canonical form (up to reordering the blocks), and the number of Jordan blocks associated to each eigenvalue is equal to the dimension of the eigenspace for that eigenvalue, the dimension of the eigenspace of 1 is bounded above by $n_1 - 1$. Since generically (away from a Lebesgue measure 0 set) $(W^1 \ b^1)$ has rank $= \min\{n_1, n_0 + 1\}$, we conclude that $n_0 + 1 \leq n_1 - 1$. But if the dimensions of the layers satisfy condition (i), then we have $n_0 \geq n_1 - 1$, so $g_1 = \text{Id}$. Applying this logic to each equation in turn, working from the top to the bottom in the list of equations above and using condition (i), we conclude that $g_\ell = \text{Id}$ for all $\ell$ and hence the stabilizer of a generic $\theta$ is trivial.

We arrive at the same conclusion by applying the same reasoning to the transpose of $g_\ell$, working from the final equation to the first. In this case, we will need to restrict to symmetric matrices $g_\ell$ in order for it to be possible for the rows of a generic $W^\ell$ to be in the 1-eigenspace of $(g_\ell)^T$ and simultaneously have a generic $b^\ell$ be in the 1–eigenspace of $g^\ell$. But if we restrict to symmetric $g_\ell$ (that is, $g_\ell = g_\ell^T$) and use assumption (ii), then the same argument as in the previous paragraph tells us that for generic $\theta$, $g_\ell = \text{Id}$ for all $\ell$. $\square$

**Definition A.9.** *For $0 \leq k \leq n$ let $\sigma_{[1:k]} : \mathbb{R}^n \to \mathbb{R}^n$ denote the function that applies the activation function ReLU(x) (resp., Id(x)) to the first $k$ components (resp., to the last $n - k$ components).*

*A* mixed ReLU-linear *neural network of architecture $(n_0, (n_1^R, n_1^L), \ldots, (n_{d-1}^R, n_{d-1}^L)|n_d)$ is a neural network of architecture $(n_0, n_1^R + n_1^I, \ldots, n_{d-1}^R + n_1^L \,|\, n_d)$ whose $\ell$th layer map is*

$$F^\ell(x) := \begin{cases} \sigma_{[1:n_\ell^R]}(W^\ell x + b^\ell) & \text{for } 1 \leq \ell < d \\ W^\ell x + b^\ell & \text{for } \ell = d. \end{cases} \tag{5}$$

**Proposition A.10.** *Given any parameter $\theta \in \Omega$ in a mixed ReLU-linear neural network of architecture*

$$(n_0, n_1^R + n_1^I, \ldots, n_{d-1}^R + n_{d-1}^L \mid n_d),$$

*all parameters in the $H$-orbit of $\theta$ for the hidden symmetry subgroup*

$$H := \{D_+(n_1^R) \times GL(n_1^L)\} \times \ldots \times \{D_+(n_{d-1}^R) \times GL(n_{d-1}^L)\}$$

*are equivalent to $\theta$. That is:*

$$F_{h\theta}(x) = F_\theta(x) \quad \forall \; x \in \mathbb{R}^{n_0}, h \in H.$$

*Moreover, if the architecture is roughly monotonic (Definition A.1), then for almost all parameters $\theta$,*

$$dim(H\theta) = dim(H) = \sum_{\ell=0}^{d-1} n_\ell^R + \sum_{\ell=1}^{d-1} (n_\ell^L)^2.$$

*Proof.* Let $D_+(n_\ell^R) \times GL(n_\ell^L) \subseteq GL(n_\ell)$ be the subgroup of $GL(n_\ell)$ consisting of 2-block matrices with an $n_\ell^R \times n_\ell^R$ upper-left block of diagonal matrices with positive entries on the diagonal, and an $n_\ell^L \times n_\ell^L$ lower-right block of invertible matrices. The subgroup

$$H = \{D_+(n_1^R) \times GL(n_1^L)\} \times \ldots \times \{D_+(n_{d-1}^R) \times GL(n_{d-1}^L)\}$$

acts on $\theta \in \Omega$ as in Equation 3. dand since $D_+(n_\ell^R)$ commutes with component-wise application of ReLU on the first $n_\ell^R$ neurons, and $GL(n_\ell^L)$ commutes with the component-wise application of the $Id$ activation function on the last $n_\ell^L$ neurons, the action of $H$ on $\Omega$ leaves the overall function invariant. Since the architecture is roughly monotonic, for almost all parameters $\theta \in \Omega$, the stabilizer of the $H$ action is trivial by Lemma A.7 and Proposition A.8, so the orbit-stabilizer theorem tells us:

$$\dim(H\theta) = \dim(H) - 0 = \sum_{\ell=1}^{d-1} n_\ell^R + \sum_{\ell=1}^{d-1} (n_\ell^L)^2.$$

$\square$

The following theorem says that if there are any neurons for a parameter $\theta$ that have fixed activation status (always-active or always-inactive) on an entire batch $X$ of data, then the dimension of the space of local data-dependent parameter space symmetries matches the dimension of the hidden equivalence group for a mixed ReLU-linear network where all of the fixed-status neurons are treated as linear.

**Theorem 1.** *Let $\theta \in \Omega$ be almost any parameter[4] in a roughly monotonic feedforward ReLU network architecture, $(n_0, \ldots, n_{d-1}|n_d)$, with parametric dimension $D$, and let $X \subseteq \mathbb{R}^{n_0}$ be a subset of the domain for which $n_\ell^{fixed}$ of the hidden neurons from layer $\ell$ are either always-active or always-inactive on all of $X$. Then the batch functional dimension of $\theta$ on $X$ is at most $D - d$, where $d = \sum_{\ell=1}^{d-1} \left(n_\ell - n_\ell^{fixed}\right) + \sum_{\ell=1}^{d-1} \left(n_\ell^{fixed}\right)^2$.*

*Proof of Theorem 1.* By the well-known description of the piecewise polynomial structure of ReLU network functions (cf. the appendices of Hanin & Rolnick (2019),Grigsby et al. (2023) and Section 2.6 of Grigsby & Lindsey (2024)), on a neighborhood of each point of $X = \{x_1, \ldots, x_m\}$, $F_\theta$ is a polynomial function in the parameters $\theta$, realized as a sum of monomials determined by the open paths at $x_i$ in the computational graph for the architecture. It follows that if we delete the stably inactive neurons and replace the ReLU activation function with the Id function on the stably active neurons of the computational graph, the parameterized ReLU neural network class looks locally on $X$ like a mixed ReLU-linear network whose ReLU neurons in layer $\ell$ are precisely those neurons whose activation status is *not* fixed on $X$. The neurons in layer $\ell$ whose activation status on $X$ is *active on all of $X$* look locally like linear neurons (i.e., we can replace ReLU with the Id activation function) and the neurons in layer $\ell$ whose activation status is *inactive on all of $X$* can be deleted from the network. Explicitly, if we list the $k_\ell := k_{\ell,+} + k_{\ell,-}$ stably active and inactive neurons on $X$ last in the $\ell$th

---

[4] $\theta$ is in the complement of a Lebesgue measure 0 set defined in the proof of Proposition A.8 in the Appendix.

layer, then we can replace the component-wise application of ReLU on the last $k_\ell$ neurons with the component-wise application of Id without impacting the overall function $F_\theta$. Accordingly, the action of the subgroup, $H := \{(D_+(n_1 - k_1) \times GL(k_1)\} \times \ldots \times \{D_+(n_{d-1}) \times GL(k_{d-1})\}$, commutes with this mixed activation function. So $H_{eq}$ contains $H$, and $\dim(H) = \sum_{\ell=1}^{d-1}(n_\ell - k_\ell) + \sum_{\ell=1}^{d-1} k_\ell^2$. The statement about the functional dimension is then an immediate consequence of Corollary 4.3 of Grigsby & Lindsey (2024) and the orbit-stabilizer theorem, once we note that there exists a non-empty open ball around $\theta$ on which an open subset of $H$ acts with trivial stabilizer. □

# B ADDITIONAL PLOTS AND EXPERIMENTAL RESULTS

## B.1 BIVARIATE DATASETS

*Bivariate synthetic data:* The bivariate functions $g_1, g_2, g_3 : [0, 1]^2 \to \mathbb{R}$, defined by

$$g_1(x, y) = 1.5 + \sin(2\pi x) + \sin(2\pi y)$$
$$g_2(x, y) = 1.5 + \sin(2\pi x) + \sin(2\pi y) + 0.5\sin(4\pi x)$$
$$g_3(x, y) = 1.5 + \sin(2\pi x) + \sin(2\pi y) + 0.5\sin(4\pi x) + 0.4\sin(4\pi y).$$

For each bivariate function $g_i$, we create a dataset $\mathcal{D}_{g_i} = \{x_j, y_j\}_{j=1}^{400} \subset \mathbb{R}^2 \times \mathbb{R}$ by sampling the domain at 400 equally spaced points (a $20 \times 20$ grid). Figure 8 in the Appendix shows the graphs and datasets for the bivariate functions $g_i$.

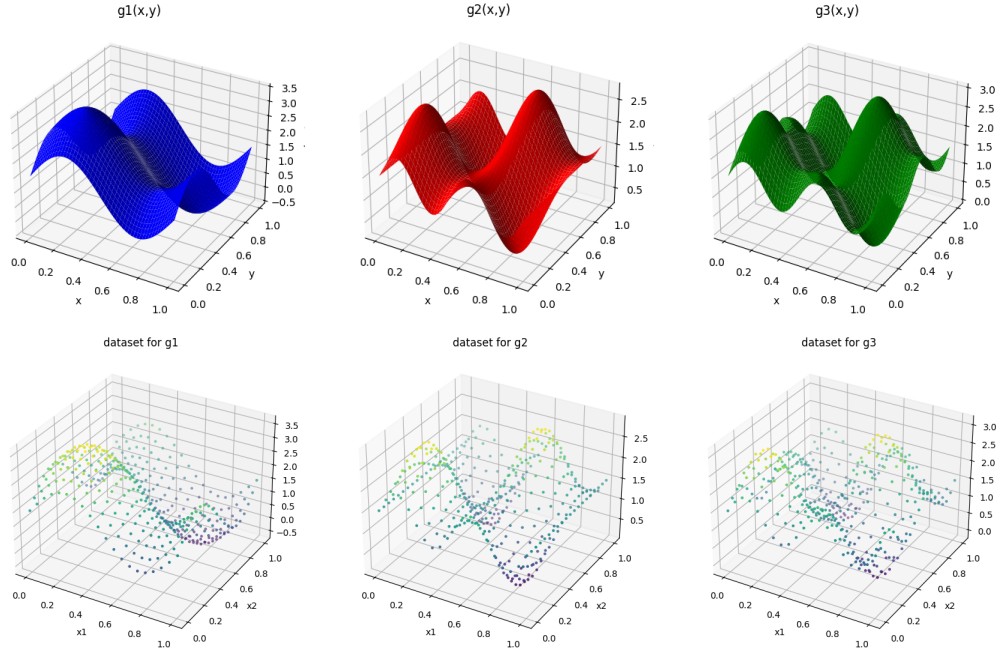

Figure 8: Graphs and datasets for the bivariate functions $g_1$, $g_2$ and $g_3$. Each datasets consists of 400 points whose inputs are uniformly distributed in the square $[0, 1]^2$.

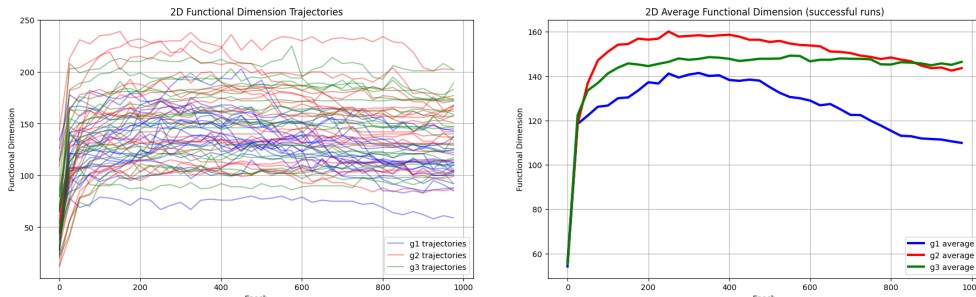

Figure 9: Evolution of functional dimension on the bivariate datasets $g_1, g_2, g_3$. Trajectories tracking the evolution of functional dimension for 20 randomly initialized training runs per dataset, with functional dimension computed every 25 epochs (left). Averages (right) .

## C  NUMBER OF LINEAR REGIONS SAMPLED BY THE DATA SET INPUTS

**Definition C.1.** *Fix a finite set $Z \in \mathbb{R}^{n_0}$. Let $\theta$ be a parameter such that every point $z_i \in Z$ is in the interior of a top-dimensional cell of the canonical polyhedral complex $\mathcal{C}(\theta)$. Then we will say that the number of linear regions of $f_\theta$ sampled by $Z$ is the number of $n_0$-dimensional cells of $\mathcal{C}(\theta)$ that have nonempty intersection with $Z$.*

*Remark* C.2. The reason for requiring that every point $z_i$ is in the interior of a top-dimensional cell is to avoid ambiguity in counting caused by a point $z_i$ being on the boundary of two or more top-dimensional cells.

Proposition C.3 is a more technical version of (and in particular implies) the statement of Proposition 4.6. By Grigsby & Lindsey (2024), for any finite batch $Z$, a full measure set of parameters $\theta$ satisfies the assumptions of Proposition C.3 .

**Proposition C.3.** *Fix an architecture $(n_0, \ldots, 1)$ of fully-connected feedforward ReLU neural networks with one-dimensional output. Fix a finite set $Z \subset \mathbb{R}^{n_0}$. Let $\theta$ be a generic, transversal, combinatorially stable parameter such that every point $z_i$ is in the interior of a $n_0$-dimensional cell of the canonical polyhedral complex $\mathcal{C}(\theta)$. Then $\dim_{ba.fun}(\theta, Z)$ is at most $(n_0 + 1)$ times the number of linear regions of $f_\theta$ sampled by $Z$.*

In particular, for an architecture with one-dimensional input and output, $\dim_{ba.fun}(\theta, Z)$ is at most twice the number of linear regions sampled by $Z$.

*Proof.* Suppose $z_1, \ldots, z_k$ are all in the interior of the same cell $C \in \mathcal{C}(\theta)$. The assumptions on $\theta$ guarantee (from results in Grigsby et al. (2022)) that there is an open neighborhood $U$ of $\theta$ on which the function $\{z_i\} \times U \to \mathbb{R}$ given by $(z, u) \mapsto f(\theta)(z)$ is affine-linear (in every coordinate). Consequently, if the point $z_{k+1}$ can be expressed as a linear combination of the points $z_1, \ldots, z_k$, then the row vector $\mathbf{J}E_{z_{k+1}}(\theta)$ can be expressed as a linear combination of the row vectors $\mathbf{J}E_{z_1}(\theta), \ldots, \mathbf{J}E_{z_k}(\theta)$. Since a top-dimensional geometric simplex in $\mathbb{R}^{n_0}$ has $n_0 + 1$ points, and batch functional dimension is the number of linearly independent rows of the corresponding matrix, the result follows. $\square$

Figures 5 and 6 demonstrate that there is a strong correlation between the number of linear regions sampled and the batch functional dimension. However, the batch functional dimension is well below the upper bound imposed by the number of linear regions as in Proposition C.3.

*Remark* C.4. Definition C.1 and Proposition C.3 are closely related to notion of decisive sets defined in Grigsby et al. (2022). A *decisive* set for a parameter $\theta$ is a set $Z \subset \mathbb{R}^{n_0}$ consisting of precisely $n_0 + 1$ points in the interior of each top-dimensional polyhedron $C \in \mathcal{C}(\theta)$ that form an $n_0$-dimensional simplex. Grigsby et al. (2022) proves that for a generic, transversal, combinatorially stable parameter, functional dimension is attained on any decisive set.

