# OpenReview forum: "Empirical NTK tracks task complexity"
_ICLR.cc/2026/Conference — Submitted to ICLR 2026_

### Official Review · Reviewer_kEaT · 2025-10-22

**Soundness:** 2
**Presentation:** 1
**Contribution:** 3
**Rating:** 4
**Confidence:** 3

**Summary:**

The paper studies the rank of the empirical NTK (eNTK) during training, both empirically and theoretically. The main contributions are:
(1) Empirical evidence showing that the eNTK computed on a batch is typically rank-deficient at the beginning of training and increases during training for several small synthetic problems.
(2) The observation that the final eNTK rank increases with task complexity and decreases with weight decay strength in the same synthetic problems.
(3) A theoretical bound on the eNTK rank for ReLU networks, formulated in terms of the number of neurons that remain always-active or always-inactive on the whole batch. The theoretical result is derived by calculating the dimension of the network’s symmetry group, thereby connecting eNTK rank deficiency to intrinsic parameter-space symmetries of ReLU architectures.

**Strengths:**

**(S1) Empirical evidence of rank deficiency in the eNTK:** The empirical results showing that the eNTK is rank-deficient at initialization are interesting, as they contradict the common assumption of full rankness often made in the NTK-regime literature.

**(S2) Relationship between symmetries of the output function and NTK rank:** The observation that the eNTK rank can be upper-bounded by the dimension of the symmetry group of the network’s output function is also interesting. While this connection is intuitive, it appears to be novel. However, it is unclear how practically useful this observation is, as discussed in the Weaknesses section.

**Weaknesses:**

**(W1) Connection between theory and experiments:** The empirical results and the theoretical bound seem largely disconnected. Although the authors show that the eNTK is rank-deficient empirically, they do not compare the theoretical bound from Thm.1 to the actual eNTK ranks observed in practice. The related issues discussed below suggest that the bound may not be predictive in the studied settings.

**(W2) Clarity of Thm.1 statement:** After reading the statement of Thm. 1, one could think that the stated upper bound may be negative with large enough $n_\ell^{\text{fixed}}$. For example, consider the architecture used in the 1D experiments, with widths $(1,15,15,15,15,1)$ and suppose $n_\ell^{\text{fixed}}=14$ in each hidden layer. Then the total number of parameters is $D=15\times 2+15\times 16\times 3+16=766$ and $d = 4\times 14\times 14+4=788$, giving $D-d<0$. Since $d$ is the orbit dimension, it cannot exceed $D$, so I suppose this case is hidden in the "almost any parameter" part of the statement (nontrivial stabilizers). However, it is not clearly stated in the theorem and does not appear in any discussion, as far as I can see.

**(W3) Is the bound in Thm. 1 mostly vacuous?** Using the same setup as in the example above but with $n_\ell^{\mathrm{fixed}}=13$, we obtain $D-d=82$, already much larger than the experimental eNTK ranks in all the provided figures. Reducing $n_\ell^{\mathrm{fixed}}$ further quickly hits the trivial upper bound on the empirical NTK rank (the “batch functional dimension” in the paper's language) given by the batch size ($100$ in the experiments). This raises the question of how often the provided bound is actually non-vacuous.

**(W4) Writing and presentation:** The paper is quite difficult to follow. Many results supporting the main claims (e.g., Figs. 7, 8, 10, 11, Proposition C.3) appear only in the appendix, while much of the main text is devoted to describing a general framework (Section 3) that is not novel, accompanied by a few very large figures. There are also editing problems, such as missing references (line 305) and incorrect appendix links (line 344).

**(W5) Scope of experiments:** The experimental scope is quite limited, and the motivation for the chosen setups is not clearly discussed. Experiments on real datasets or with more varied task structure and complexity would strengthen the results.

Overall, while the paper contains an interesting idea, the current version is not sufficiently convincing in my opinion. Therefore, I am leaning toward rejection. My assessment could improve if the authors respond to the concerns regarding the main theoretical bound, its relationship to the experiments, and the experimental scope.

**Questions:**

- What is the relationship between the main result (Thm.1) and the “roughly monotonic” assumption (A.1)?
- Did the authors compute the actual values of the bound in Thm.1 for their experimental settings?
- How do the results connect to prior work that typically observes a reduction in *effective rank* of the eNTK at the start of training? See, for example, experiments in Baratin et al. (2021) [1].

## References

[1] Baratin, Aristide, et al. “Implicit regularization via neural feature alignment.” *International Conference on Artificial Intelligence and Statistics (AISTATS)*, PMLR, 2021.

---

> ### Author Response · Authors · 2025-11-21
>
> Thank you to Reviewer kEaT for your very helpful feedback and comments. We respond to your main concerns and questions one by one below.

---

> ### Author Response · Authors · 2025-11-21
>
> * After reading the statement of Thm. 1, one could think that the stated upper bound may be negative with large enough $n_\ell^{fixed}$. For example, consider the architecture $(1,15,15,15,15,1)$ used in the 1D experiments, with widths
>  and suppose $n_\ell^{fixed} = 14$ in each hidden layer. Then the total number of parameters is $D = 766$ and $d=788$, giving $D-d$ is negative. Since $d$ is the orbit dimension, it cannot exceed $D$, so I suppose this case is hidden in the "almost any parameter" part of the statement (nontrivial stabilizers). However, it is not clearly stated in the theorem and does not appear in any discussion, as far as I can see.
>
> Thank you very much to the reviewer for this insightful comment. Indeed, we mistakenly omitted the “roughly monotonic” assumption from the statement of Theorem 1 when we prepared the paper for submission. (It did appear in the statement of Proposition A.10 in the appendix, which is the technical result upon which Theorem 1 rests.) Indeed, for the non-roughly-monotonic architecture $(1,15,15,15,15,1)$ with 14 fixed-status neurons in each hidden layer, there will be a high-dimensional non-trivial stabilizer, and hence the dimension “d” of the orbit will be much lower than the value stated in the theorem. We corrected the statement of Theorem 1 in the main body of the paper, moved the definition of roughly monotonic into Section 3 from the appendix and added a bit more detail there about the architecture restriction. This also addresses the reviewer's later question:
>
> * What is the relationship between the main result (Thm.1) and the “roughly monotonic” assumption (A.1)?

---

> ### Author Response · Authors · 2025-11-21
>
> * Is the bound in Thm. 1 mostly vacuous? Using the same setup as in the example above but with $n_\ell^{fixed} = 13$, we obtain $D - d = 82$, already much larger than the experimental eNTK ranks in all the provided figures. Reducing
>  further quickly hits the trivial upper bound on the empirical NTK rank (the “batch functional dimension” in the paper's language) given by the batch size (100 in the experiments). This raises the question of how often the provided bound is actually non-vacuous.
>
> Thank you very much to the reviewer for this insightful comment. The bound we give is only valid for roughly-monotonic architectures. Since our experiments were run on non-roughly-monotonic architectures like $(\mbox{input dimension},10,10,10,10,1)$, this theorem unfortunately cannot directly explain our empirical findings.
>
> However, the bound for roughly-monotonic architectures in general is novel and nonvacuous, especially for larger data sets. It is the first example in the literature of a complete calculation of an upper bound on the dimension of a hidden equivalence group for ReLU networks for an infinite family of architectures.
>
> An example of a roughly-monotonic architecture of univariate functions is (1,5,4,3,2,1). In this case,
> $D = 2*5+6*4+5*3+4*2+3*1$.
>
> If all but 1 of the neurons in each hidden layer has fixed activation status, then
> $d = (1+4^2) + (1 + 3^2) + (1 + 2^2) + (1 +1^2) = 34$. This gives an upper bound on functional dimension of $60-34=26$; much better than the previously-established upper bound of $60-(5+4+3+2)=46$.

---

> > ### Author Response · Authors · 2025-11-21
> >
> > * Writing and presentation: The paper is quite difficult to follow. Many results supporting the main claims (e.g., Figs. 7, 8, 10, 11, Proposition C.3) appear only in the appendix, while much of the main text is devoted to describing a general framework (Section 3) that is not novel, accompanied by a few very large figures. There are also editing problems, such as missing references (line 305) and incorrect appendix links (line 344).
> >
> > Thank you to the reviewer for this feedback. We fixed the missing reference and incorrect appendix links. We streamlined Section 3 as much as possible and moved Prop. C.3 back into the main body of the paper.

---

> > > ### Author Response · Authors · 2025-11-21
> > >
> > > * Scope of experiments: The experimental scope is quite limited, and the motivation for the chosen setups is not clearly discussed. Experiments on real datasets or with more varied task structure and complexity would strengthen the results.
> > >
> > > Thank you for this feedback.  We added two types of additional experiments in the main body of the paper: 1) we replaced our noiseless univariate synthetic experiments with noisy synthetic data and 2) we trained equivalent network architectures on two regression tasks of increasing complexity using a real-world data set (the UCI energy efficiency data set). All of the central findings are corroborated on these data sets. See Figures 1-3 and Sections 5 and 6 of the revised rebuttal draft.

---

> > > > ### Author Response · Authors · 2025-11-21
> > > >
> > > > * Did the authors compute the actual values of the bound in Thm.1 for their experimental settings?
> > > >
> > > > Thank you so much for this question. As noted in the responses to the points above, Theorem 1 does not apply in our experimental setting. We tried adjusting the architecture in our experiments to be roughly monotonic, but the networks unfortunately did not train effectively, and the computation of the stabilizer in the non-roughly-monotonic setting is beyond the scope of what we can do in a rush.

---

> > > > > ### Author Response · Authors · 2025-11-21
> > > > >
> > > > > * How do the results connect to prior work that typically observes a reduction in effective rank of the eNTK at the start of training? See, for example, experiments in Baratin et al. (2021).
> > > > >
> > > > > Thank you so much to the reviewer for making us aware of this very relevant related work. The effective rank of the eNTK, as measured in that paper, should be closely related to the algebraic notion of rank we measure in our experiments, but the quantities are not exactly the same, so it is not surprising that their behavior during training differs. Note that the rank we measure is defined to be the number of singular values above a default tolerance value (see Section “Batch functional dimension computation” in our paper for details). We added this paper to our bibliography and commented on it in the related work section. We are especially intrigued by the experiments in that paper showing that the central kernel alignment (CKA) score of the eNTK with the class label kernel increases during training. We don’t yet know how to explain this - it is an interesting future research question.

---

> > > > > > ### Comment · Reviewer_kEaT · 2025-11-26
> > > > > >
> > > > > > Thank you for the detailed response. I appreciate that the authors have addressed the missing assumption issue in the main theorem. However, the necessity of this assumption makes the disconnect between the theoretical results and the experiments (W1) even more evident. Since the authors explicitly state that they were unable to design meaningful experiments in the roughly-monotonic setting, this raises further concerns about the applicability of the theoretical results.
> > > > > >
> > > > > > In my view, for this paper to be suitable for acceptance, it would need to either (i) provide a coherent theory accompanied by empirical validation in the settings covered by the theory (even if those settings are less practically relevant) or (ii) present extensive experiments on realistic architectures and datasets, including at least standard deep learning benchmarks such as MNIST or CIFAR. As neither of these is currently satisfied, I am keeping my score.

---

### Official Review · Reviewer_jdWV · 2025-10-27

**Soundness:** 2
**Presentation:** 2
**Contribution:** 2
**Rating:** 2
**Confidence:** 4

**Summary:**

This paper analyzes the empirical NTK rank and its change during training. Claims are made based on two experiments on simple synthetic data and a simple model. A bit theory is developed on the symmetry of the network, but lacks connection with the paper’s major goal.

**Strengths:**

n/a

**Weaknesses:**

1: A connection of the theory of the paper, simply Theorem 1, and the major claims of the paper is missing. The paper mainly claims about the rank of empirical NTK and its change during training. However, (a) Theorem 1 seems to be independent of the training procedure. (b) what is its implication on the NTK rank?


2: The presentation of the paper can be largely improved, especially in Section 4.1.
In most of this section, the paper is just filled with a few technical mathematical concepts which have no clear connection to the paper’s context. This makes it hard to understand the discussion, even though I have the knowledge of these math concepts.

The connection to the paper’s context is made clear only in Remark 4.3 (quite near the end of the section), where permutation and scaling actions are discussed. I believe a better way is to (1) put the permutation and scaling actions early on (also not under a remark), and (2) put the technical concepts, such as Lie Group, orbits etc, into appendix.

In addition, given the symmetry (permutation and scaling) is quite limited and simple, I doubt the necessity of introducing the concept of Lie groups.

3: As an empirical work, this paper only experimented on extremely simple synthetic data, which are far from enough to be conclusive. I expect verifications of the empirical claims across several real-world datasets and network architectures.

**Questions:**

no further questions

---

> ### Author Response · Authors · 2025-11-21
>
> Thank you to Reviewer jdWV for your very helpful feedback. We respond to your main concerns and questions one by one below.

---

> > ### Author Response · Authors · 2025-11-21
> >
> > * A connection of the theory of the paper, simply Theorem 1, and the major claims of the paper is missing. The paper mainly claims about the rank of empirical NTK and its change during training. However, (a) Theorem 1 seems to be independent of the training procedure. (b) what is its implication on the NTK rank?
> >
> > The novel contributions of the paper are 1) an empirical result about a relationship between evolution of the rank of the empirical NTK and task complexity, along with 2) an explicit theoretical connection between empirical NTK and parameter space symmetries, and 3) two theoretical results partially explaining the low rank at initialization. We do not yet have a theoretical explanation for the evolution of functional dimension during training, or an understanding of how the particular optimization procedure interacts with functional dimension. This is a very interesting research question, and one we will pursue in future projects. We revised the organization and text in Section 4 substantially to make the connection between empirical and theoretical results more clear.

---

> > > ### Author Response · Authors · 2025-11-21
> > >
> > > * The presentation of the paper can be largely improved, especially in Section 4.1. In most of this section, the paper is just filled with a few technical mathematical concepts which have no clear connection to the paper’s context. This makes it hard to understand the discussion, even though I have the knowledge of these math concepts.
> > > * The connection to the paper’s context is made clear only in Remark 4.3 (quite near the end of the section), where permutation and scaling actions are discussed. I believe a better way is to (1) put the permutation and scaling actions early on (also not under a remark), and (2) put the technical concepts, such as Lie Group, orbits etc, into appendix.
> > > * In addition, given the symmetry (permutation and scaling) is quite limited and simple, I doubt the necessity of introducing the concept of Lie groups.
> > >
> > > Thank you very much for this helpful feedback. We understand that the permutation and scaling invariance of ReLU networks is straightforward and well-known to experts, so we followed your suggestion to state that first in order to orient the reader. Mentioning Lie groups and their actions is important to our theoretical results, since it makes clear how certain less-well-known data-dependent symmetries can arise, leading to larger-dimensional level sets of the empirical loss. Also, it connects these ideas to a vast mathematical literature that has been developed over more than a century. Because of this history, many results that would be difficult to prove from scratch are much easier in this setting. We substantially revised and reorganized Section 4 in a way that we hope makes the main results more clear and accessible.

---

> > > > ### Author Response · Authors · 2025-11-21
> > > >
> > > > * As an empirical work, this paper only experimented on extremely simple synthetic data, which are far from enough to be conclusive. I expect verifications of the empirical claims across several real-world datasets and network architectures.
> > > >
> > > > Thank you for this feedback.  We added two types of additional experiments in the main body of the paper: 1) we replaced our noiseless univariate synthetic experiments with noisy synthetic data and 2) we trained equivalent network architectures on two regression tasks of increasing complexity using a real-world data set (the UCI energy efficiency data set). All of the central findings are corroborated on these data sets, see Figures 1-3 and Sections 5 and 6.

---

### Official Review · Reviewer_B9ZF · 2025-10-31

**Soundness:** 3
**Presentation:** 3
**Contribution:** 2
**Rating:** 4
**Confidence:** 4

**Summary:**

This paper investigates the empirical Neural Tangent Kernel (NTK) in finite-width deep ReLU networks, focusing on its relationship with task complexity and training dynamics. The authors show that, contrary to common belief, the empirical NTK is typically rank-deficient at initialization. They demonstrate that its rank increases during training until function stabilization, correlates positively with dataset complexity, and is suppressed by stronger weight decay. Theoretically, they link low functional dimension to data-dependent parameter-space symmetries and provide an upper bound based on the number of linear regions sampled by the data. These findings offer new insights into optimization biases and implicit regularization in deep ReLU networks.

**Strengths:**

1. The paper presents a highly original and significant empirical finding by demonstrating that the empirical NTK for deep, finite-width ReLU networks is consistently rank-deficient at initialization, contrary to conventional wisdom.

2. The work provides creatively linking the NTK's rank to the novel concept of data-dependent parameter-space symmetries and providing an upper bound based on linear regions, it offers a compelling and elegant mechanistic explanation for the observed low-rank phenomenon.

3. The paper achieves remarkable clarity by grounding abstract concepts like the NTK and functional dimension in concrete, measurable quantities (matrix rank, linear region count).

**Weaknesses:**

1.  The experimental validation is conducted exclusively on small, synthetic, noiseless datasets (univariate and bivariate functions). This leaves it unclear whether the central finding—that the NTK rank tracks task complexity—generalizes to high-dimensional, noisy, real-world data (e.g., CIFAR or ImageNet), where the notion of "complexity" is less well-defined.

2. While the paper provides a compelling theoretical explanation (data-dependent symmetries from fixed activation patterns), it lacks direct empirical measurement to substantiate this as the primary cause. Quantifying the number of "always-active/inactive" neurons throughout training and directly correlating this count with the observed drops in functional dimension would significantly strengthen the theoretical results.

3. Proposition C.3 establishes an upper bound for functional dimension based on the number of linear regions, but the results show the actual dimension is "well below" this bound. The paper does not sufficiently investigate what factors determine this gap.

**Questions:**

1. Your theoretical framework and experiments are compelling on low-dimensional synthetic data. Could you provide evidence or discuss whether the strong correlation between empirical NTK rank and "task complexity" holds for higher-dimensional, real-world datasets (e.g., CIFAR, ImageNet )? In such domains, how would you propose to define or measure "task complexity" independently to validate this relationship?

2. You propose that data-dependent symmetries (from always-active/inactive neurons) are a primary cause for the low-rank NTK. Did you track the proportion of such neurons throughout training? Could you show that a drop in functional dimension coincides with an increase in neurons achieving a fixed activation status?

---

> ### Author Response · Authors · 2025-11-21
>
> Thank you to Reviewer B9ZF for your very helpful feedback. We respond to your main concerns and questions one by one below.

---

> ### Author Response · Authors · 2025-11-21
>
> * The experimental validation is conducted exclusively on small, synthetic, noiseless datasets (univariate and bivariate functions). This leaves it unclear whether the central finding—that the NTK rank tracks task complexity—generalizes to high-dimensional, noisy, real-world data (e.g., CIFAR or ImageNet), where the notion of "complexity" is less well-defined.
>
> Thank you to the reviewer for this feedback. We added two types of additional experiments in the main body of the paper: 1) we replaced our noiseless univariate synthetic experiments with noisy synthetic data (See our new Figures and 2) we trained equivalent network architectures on two regression tasks of differing complexity using a real-world data set (the UCI energy efficiency data set). All of the central findings are corroborated on these data sets. See Figures 1-3 and Sections 5 and 6 in the revised rebuttal draft.

---

> ### Author Response · Authors · 2025-11-21
>
> * While the paper provides a compelling theoretical explanation (data-dependent symmetries from fixed activation patterns), it lacks direct empirical measurement to substantiate this as the primary cause. Quantifying the number of "always-active/inactive" neurons throughout training and directly correlating this count with the observed drops in functional dimension would significantly strengthen the theoretical results.
>
> Thank you to the reviewer for this excellent point. Running this type of experiment in time to revise the paper for the current submission is not possible, so we adjusted the text towards the beginning of Section 4 indicating that the theoretical mechanisms we propose may not fully explain the empirical phenomena.
>
> *  Proposition C.3 establishes an upper bound for functional dimension based on the number of linear regions, but the results show the actual dimension is "well below" this bound. The paper does not sufficiently investigate what factors determine this gap.
>
> We have a similar response to this point as to the previous one. We do not mean to imply that our theory gives a full explanation for the empirical phenomena we observe - merely that it gives a concrete description of the effect on functional dimension of certain mechanisms. See the adjusted text towards the beginning of Section 4.

---

> > ### Author Response · Authors · 2025-11-21
> >
> > * Your theoretical framework and experiments are compelling on low-dimensional synthetic data. Could you provide evidence or discuss whether the strong correlation between empirical NTK rank and "task complexity" holds for higher-dimensional, real-world datasets (e.g., CIFAR, ImageNet )? In such domains, how would you propose to define or measure "task complexity" independently to validate this relationship?
> >
> > Thank you for this excellent question and suggestion. We trained equivalent network architectures on two regression tasks of increasing complexity using a small real-world data set (the UCI energy efficiency data set). All of the central findings are corroborated on this data set. Since functional dimension calculations during training can be expensive, and we wanted to use an architecture similar to the one we used for our synthetic data set, we chose a widely-used data set with relatively low-dimensional (8-dimensional) input space, as opposed to a larger data set like CIFAR or ImageNet. We incorporated our empirical results into the rebuttal draft of our manuscript. See Figure 3, and Section 5 describing the data set.

---

> > > ### Author Response · Authors · 2025-11-21
> > >
> > > * You propose that data-dependent symmetries (from always-active/inactive neurons) are a primary cause for the low-rank NTK. Did you track the proportion of such neurons throughout training? Could you show that a drop in functional dimension coincides with an increase in neurons achieving a fixed activation status?
> > >
> > > Thank you for this excellent point. Running this type of experiment in time to revise the paper for the current submission is not possible, so we adjusted the text indicating that the theoretical mechanisms we propose are one of several possible mechanisms and therefore may not fully explain the empirical phenomena.

---

### Official Review · Reviewer_c9iJ · 2025-10-31

**Soundness:** 3
**Presentation:** 4
**Contribution:** 3
**Rating:** 4
**Confidence:** 2

**Summary:**

The paper investigates the empirical behavior of Neural Tangent Kernels (NTKs) across various ML algorithms, with an emphasis on understanding how NTK dynamics reflect or correlate with the underlying complexity of the algorithms. The authors provide a comprehensive experimental and analytical study on the evolution of the empirical NTK during training, examining its relation to key measures of task difficulty.

**Strengths:**

- The paper shows extensive experimental evaluations.

- The conceptual innovation of functional dimension as a data and task dependent metric is significant.

**Weaknesses:**

A few important discussions with existing literature are missing.

**Questions:**

How robust is the correspondence between functional dimension, as defined via the NTK spectrum, and the capacity of a network to generalize?

Can you compare your results with the Scaling neural tangent kernels via sketching and random features?

What are examples of nonstandard data distributions for which the analysis in 'Fine-Grained Analysis of Optimization and Generalization for Overparameterized Two-Layer Neural Networks' fails to accurately capture the behavior of finite-width networks?

---

> ### Author Response · Authors · 2025-11-21
>
> Thank you to Reviewer c9iJ for your very helpful feedback. We respond to your main concerns and questions one by one below.

---

> > ### Author Response · Authors · 2025-11-21
> >
> > *  "A few important discussions with existing literature are missing."
> >
> > We added a few references in our bibliography that address your questions as well as the questions of Reviewer kEaT and commented on them in an expanded related work section.

---

> > > ### Author Response · Authors · 2025-11-21
> > >
> > > * "How robust is the correspondence between functional dimension, as defined via the NTK spectrum, and the capacity of a network to generalize?"
> > >
> > > Thank you to the reviewer for this question. There are only partial theoretical answers in the literature. For example, in _Spectral Bias and Task-Model Alignment Explain Generalization in Kernel Regression and Infinitely Wide Neural Networks_ (Canatar, Bordelon, Pehlevan 2020) and _Just Interpolate: Kernel ridgeless regression can generalize_ (Liang, Rakhlin 2021), the authors establish generalization bounds from the effective dimension of the empirical NTK (which measures the rapidity of spectral decay). These results are only for networks in the infinite width limit. There is a recent paper, _On functional dimension and persistent pseudodimension_ (Grigsby, Lindsey 2024), that conjectures that functional dimension for finite width networks is equal to a local measure of complexity (persistent pseudodimension) from which generalization bounds should be extractable.

---

> > > > ### Author Response · Authors · 2025-11-21
> > > >
> > > > * Can you compare your results with the Scaling neural tangent kernels via sketching and random features?
> > > >
> > > > Our understanding is that the literature on scaling NTKs and random features are relevant only for kernel regression, i.e. for neural networks evolving via gradient flow/descent in the infinite-width limit. This is not the regime in which we are operating in our experiments or theory. Can the reviewer provide some more context or specific references we can read to compare with what we are doing in our work?

---

> > > > > ### Author Response · Authors · 2025-11-21
> > > > >
> > > > > * What are examples of nonstandard data distributions for which the analysis in 'Fine-Grained Analysis of Optimization and Generalization for Overparameterized Two-Layer Neural Networks' fails to accurately capture the behavior of finite-width networks?
> > > > >
> > > > > The original sentence, “...especially in the presence of nonstandard data distributions” that followed the reference to the above work in our original version was not intended to explicitly point to that work. In fact, the assumptions on the data distribution in the referenced paper are very mild, as noted by the reviewer. To avoid confusion, we deleted that phrase from the current draft.

---

### Meta-Review · Area_Chair_XRYv · 2026-01-04

**Summary:**

This paper empirically and theoretically studies the Neural Tangent Kernel (NTK) of finite-width deep ReLU networks, showing that the empirical NTK is typically rank-deficient at initialization and that its rank evolves during training in ways correlated with task complexity and regularization.
While the ideas are interesting, all reviewers have reached a consensus that the current version requires further strengthening, particularly in empirical validation, contextualization with prior work, and clarity of presentation, to support a more significant and convincing contribution.

**Reviewer Concerns:**

- Reviewer **c9iJ** asked for: (1) discussion and comparison with previous theoretical and empirical work. I think that some of these points were addressed during the rebuttal, while others were not.
- Reviewer **B9ZF** asked for: (1) stronger empirical results, and (2) further clarification of the results and the significance of the contribution. Although the authors provided additional empirical results (on the UCI Energy Efficiency dataset), I do **not** find them sufficiently convincing.
- Reviewer **jdWV** asked for: (1) clarification of the results and the significance of the contribution, (2) improved presentation of the results, and (3) stronger empirical validation. I think that most of these concerns remain outstanding.
- Reviewer **kEaT** raised concerns largely overlapping with those of Reviewer jdWV, and I believe that most of these issues are still outstanding.

**Reviewer Scores:**

I do not believe that Reviewers **c9iJ**, **B9ZF**, or **kEaT** would have increased their scores.

It is possible that Reviewer **jdWV** might have increased their score, but only very slightly (e.g., from 2 to 4).

---

### Decision · Program_Chairs · 2026-01-26

Reject